# A Personalized Approach to Radical Cystectomy Can Decrease Its Complication Rates

**DOI:** 10.3390/jpm12020281

**Published:** 2022-02-14

**Authors:** Przemyslaw Adamczyk, Pawel Poblocki, Cyprian Michalik, Mateusz Kadlubowski, Jan Adamowicz, Witold Mikolajczak, Tomasz Drewa, Kajetan Juszczak

**Affiliations:** 1Department of General and Oncologic Urology, Nicolaus Copernicus Hospital, 87-100 Torun, Poland; poblocki@gmail.com (P.P.); kadlub1@o2.pl (M.K.); witoldmikolajczak18@gmail.com (W.M.); t.drewa@wp.pl (T.D.); 2Department of Oncological Urology, Maria Sklodowska-Curie Memorial Cancer Center and Institute of Oncology, 31-115 Krakow, Poland; cyp2d8@gmail.com; 3Clinic of General and Oncologic Urology, Collegium Medicum of Nicolaus Copernicus University, 87-134 Bydgoszcz, Poland; adamowicz.jz@gmail.com (J.A.); kaj.juszczak@gmail.com (K.J.)

**Keywords:** bladder cancer, radical cystectomy, complications, ASA scale, CCI scale, ECOG scale, G-8, Clavien–Dindo scale

## Abstract

The aim of this study was to assess the influence of a patient’s general status on perioperative morbidity and mortality after radical cystectomy, and to assess which of the used scales is best for the prediction of major complications. The data of 331 patients with muscle-invasive bladder cancer, who underwent radical cystectomy, were analyzed. The general status was assessed according to the American Society of Anesthesiologists (ASA), Charlson Comorbidity Index (CCI), Eastern Cooperative Oncology Group (ECOG), and Geriatric-8 (G-8) scales. Complications were classified according to the Clavien–Dindo classification system. In a group of patients with the highest complication rate according to the Clavien–Dindo scale, (i) statistically more patients rated high according to the ASA and ECOG scales, (ii) patients had significantly higher CCI scores (minor complications (I-II), and (iii) there were significantly more patients rated as frail with G8—predominantly those with 11 points or fewer in the scale. A patient’s general status should be assessed before the start of therapy because patients with a high risk of death or serious complications (evaluated with any rating scale) should be offered conservative treatment. None of the scales can describe the risk of cystectomy, because the percentage of patients with major complications among those who achieved worse score results on any scale was not significantly different from the percentage of patients with major complications in the general group.

## 1. Introduction

Bladder cancer is the fourth most common cancer in men and the twelfth most common cancer in women [1]. At presentation, around 30% of cases are muscle-invasive bladder cancer. Radical cystectomy (RC) with neoadjuvant or adjuvant chemotherapy remains the standard treatment in these cases [2], performed as an open (ORC), laparoscopic (LRC), or robot-assisted surgery (RARC) with ileal conduit (IC), orthotopic neobladder (ON), or simple ureterocutaneostomy (UCS) as the urinary diversion. The mortality rates among patients who undergo operations are estimated at 1.2–32% after 30 days, and 2.3–8.0% after three months in the case of RARC. Early postoperative complications (<90 days after surgery) were reported in 58% of patients, which can be linked to the type of urinary diversion [3]. However, the risk of major complications of radical cystectomy is greater in older patients due to frequent comorbidities and general frailty, but considerable improvements in perioperative management and the use of minimally invasive procedures offer a favorable risk-benefit profile for elderly adults who were previously disqualified from surgery [4,5].

Until recently, only age and standard clinical assessment were used to guide treatment decisions, but several scoring systems have been developed to recognize patients’ general status before RC. From those, Geriatric-8 (G8), the Eastern Cooperative Oncology Group (ECOG) scale, and the Charlson Comorbidity Index (CCI) are commonly used. To assess the risk of anesthesia prior to surgery, the American Association of Anesthesiology (ASA) scale is usually applied [6,7,8,9].

Platinum-based chemotherapy, which is commonly used in bladder cancer patients, is effective, but its use is limited by its severe, dose-limiting side effects [10]. Therefore, the use of those agents is personalized and usually given to younger patients with favorable prognoses [10]. 

The personalized approach also depends on the histopathological result of tissue taken during transurethral resection of the tumor before radical treatment. The majority of bladder cancers are classified as pure urothelial carcinoma, and the rest are assessed as other histological variants [11], including adenocarcinoma, which usually derives from bladder urachus. Several patterns of this cancer exist, and differentiating urachal and non-urachal subtypes of adenocarcinoma is essential in the personalized approach to the management of this disease because partial cystectomy can be performed in patients with pure urachal adenocarcinoma [12]. The same applies to the treatment of patients with other histopathological variants. Bladder SCC is usually treated with palliative chemotherapy based on neuroendocrine-type regimens using a platinum drug (cisplatin in healthy patients) because it is usually found in the advanced stage [13].

Besides histology, a personalized approach should also include assessment of general and nutritional status, because low albumin levels can have a negative impact on the risk of perioperative morbidity, and hypoalbuminemia is a recognized risk factor of postoperative complications (e.g., surgical wound healing and/or gastrointestinal complications) and infection [14]. However, the precise impact of patients’ nutritional status or clinical factors on the complication rates after RC remain unclear [15].

The patient’s general status should be assessed before surgery, using specially designed scales to adjust the type of treatment to the patient’s fragility status. The most common is the ECOG scale. Another scale is the CCI, which is an independent risk factor for postoperative mortality [7]. It considers 19 general conditions, and the results are used to calculate the Charlson Probability Index of 10-year mortality risk. 

The G8 is a simple eight-item screening scale, which considers seven factors from the Mini Nutritional Assessment tool [16]. It was developed to identify healthy older adult cancer patients who can receive standard therapy. The G8 may also have predictive value for patient outcomes after radical cystectomy. Even in patients with high scores, and in whom a full geriatric assessment may not yet be indicated, G8 might still predict poor outcomes [17]. A previous study revealed in multivariable analysis that patients with neoplasmatic disease with a low G8 score had significantly shorter overall survival [18].

This study aimed to explore the influence of the patient’s general status, measured by four designed scales (ASA, G8, ECOG, and CCI), on perioperative morbidity and mortality related to RC. We also assessed which of the four scales is best in terms of prediction of major complications after RC. 

## 2. Materials and Methods

### 2.1. Study Design and Patients

This retrospective study included data from 331 consecutive patients from three institutions who underwent RC between 2013 and 2021. Of these, 80 underwent operations by ORC, 61 by LRC, and 190 by RARC. The choice of surgical technique depended on availability in the department and the surgeon’s decision. Indications for RC were in accordance with the guideline of the European Association of Urology (EAU) [19]. Neoadjuvant chemotherapy was administered according to the decision of the multidisciplinary team. During RC (which included resection of the prostate in men and the reproductive system in women), the obturator, external, internal, common iliac, and presacral lymph nodes were dissected.

Surgery was performed by four different surgeons, but most laparoscopic and all robot-assisted surgeries were performed by the same operator. All consecutive patients that qualified for surgery in one center were operated by the same method, irrespective of any clinical variables (age, general status, disease stage, etc.).

### 2.2. Data Collection

All patients underwent a preoperative examination, including routine laboratory tests, a chest radiogram, an abdominal ultrasonography scan, a computed tomography (CT) scan, and/or magnetic resonance imaging (MRI). Oncological variables and results were noted, and neoplasm staging was performed according to the TNM classification system [20].

Anesthesia risk was assessed and scored according to the ASA physical status classification system. General status was assessed according to the CCI, ECOG, and G8 scales. Complications were classified according to the Clavien–Dindo classification system. Major complications were defined as a Clavien–Dindo score of grades 3–4, and minor as grades 1–2 [21]. Groups were divided into subgroups according to the method of urinary diversion: ileal conduit, orthotopic neobladder, and simple UCS. Complications were stratified according to urinary deviation.

### 2.3. Ethical Considerations

Because this study was a retrospective chart review, informed consent was not required. All procedures were performed in accordance with the ethical standards of the Ethics Committee of the Nicolaus Copernicus University (number 439/2013) and with the Declaration of Helsinki (1964) and its later amendments or comparable ethical standards.

### 2.4. Statistical Analysis

The normality of the data was checked by the Shapiro–Wilk test. The chi-square test was used to identify associations between dichotomous and categorical data. The Kruskal–Wallis test was used to compare the patients’ results among the categories followed by the post hoc Dunn multiple comparisons test. Statistical significance was considered at *p* < 0.05 for all tests.

## 3. Results

Women accounted for 20.8% of the study population. The median age was 68.5 (range 17–91) years (Q1–Q3: 62.0–74.0), the mean age 67.82 (SD 9.32).

### 3.1. Frequency of ASA, ECOG, G8, and CCI

Patients from the study group were scored according to the ASA, ECOG, and G8 scales. The details are presented in Table 1.

### 3.2. Ten-Year Survival According to the CCI

It was observed that according to the CCI, 106 patients (32%) had no chance of 10-year survival. In the rest of the group, the chance of 10-year survival was between 2% and 98% (Figure 1).

### 3.3. Complication Rates in the Study Group and Subgroups According to the Type of Urinary Diversion

In the whole group, the most frequent were Clavien–Dindo grades 1 and 2 (82% of patients), and the grades were dependent on the type of urinary diversion; when more complex urinary derivation is applied, a significantly higher percentage of patients suffer higher grades of complications (Figure 2).

### 3.4. Evaluation of the Relationship between Preoperative Status and Number of Complications

#### 3.4.1. ASA Scale vs. Clavien–Dindo Scale

The percentage of minor complications was highest in patients with ASA 1 and lowest in those with ASA 4. Death (Clavien–Dindo grade 5) was only noted in patients assessed as ASA 4 and ASA 3 (Figure 3).

#### 3.4.2. ECOG Scale vs. Clavien–Dindo Scale

In the group of patients with the highest complication rate according to the Clavien–Dindo scale, there were statistically more patients rated high according to the ECOG scale (*p* = 0.021) (Figure 4).

At the same time, there were no significant differences in subgroups according to the type of urinary diversion (*p* > 0.05).

#### 3.4.3. CCI Scale vs. Clavien–Dindo Scale

Patients with the highest complication rate according to the Clavien–Dindo scale had significantly higher CCI scores (H = 9.41, *p* < 0.01 vs. minor complications (1–2)—*p* < 0.01 and vs. major complications (3–4) *p* < 0.05) (Figure 5). In relation to the urinary derivation, the study group was divided according to the CCI level. In the group of patients with Clavien–Dindo 5 complications, there were significantly more patients with higher CCI scores (this applies to patients with UCS *p* = 0.0134; in the ileal conduit and Studer neobladder group, there were no such complications).

#### 3.4.4. G8 Scale vs. Clavien–Dindo Scale

In the group with Clavien–Dindo grade 5, there were significantly more patients rated as frail with the G8 scale, predominantly those with 11 points or fewer in the scale, whereas in the group of patients with minor and major complications, there were predominantly those with a cut-off level above 14 (H = 17.35, *p* < 0.01; comparison vs. minor and vs. major *p* < 0.001) (Figure 6). When divided into groups according to the urinary derivation, the relationship is similar. In the UCS group with Clavien–Dindo grade 5 complications, patients with G8 score of 11 and lower—*p* < 0.001 dominated (in IC and SN, there were no Clavien–Dindo grade 5 complications). 

### 3.5. Co-Morbidities vs. Complication Rate

The study group was evaluated according to the nourishing status (measured by protein concentration, renal insufficiency, and diabetes in relation to the type of urine derivation, with no significant difference between study groups (respectively, *p* = 0.233, *p* = 0.404, and *p* = 0.165)). The same applies to the subgroups according to the urinary derivation (for USC, IL, and SN, respectively, *p* = 0.067 *p* = 0.753, and *p* = 0.891).

## 4. Discussion

Radical cystectomy in patients with advanced urothelial carcinoma combined with pre- or postoperative chemotherapy provides good outcomes regarding short-term survival; however, around two-thirds of patients suffer one or more complications within 90 days of surgery [22]. A significant group of patients who require aggressive oncological treatment are older adults, who are seen more frequently now than in previous years. Bladder cancer can be considered an age-dependent neoplasm because prolonged exposure to carcinogenic agents for many years makes it more frequent in older age. A personalized approach to the surgical treatment of such patients must be adapted not only to the cancer histology but also to the stage of the disease and the patient’s general status.

The current, classical approach to radical cystectomy, based on histology and imaging studies, can be misleading. During pre-operative staging, tumors classified on a pathological basis as low risk show highly invasive biological behavior, with a high risk of local progression and metastasis. Additionally, from observational studies, it can be seen that some tumors respond strongly to intravesical Bacillus Calmette–Guerin (BCG) therapy, while others with similar pathological grading are BCG refractory. Some tumors respond well to chemotherapy, and some are not chemosensitive. This suggests that urothelial carcinoma cannot be distinguished only on the basis of pathological examination, and outcomes of treatment cannot be predicted with the traditional classification system. The currently used pathological parameters of the tumor cannot fully reflect the true biological characteristics of bladder cancer. Therefore, a personalized approach to treatment according to the new findings of cancer biology and genetics is necessary to improve the outcomes of therapy and to distinguish those patients who require aggressive treatment. 

Current findings, based on transcriptional analysis, assign bladder cancer to one of two main molecular types: luminal and basal. With gene expression profiling, six subtypes were distinguished: luminal papillary, luminal non-specified, luminal unstable, stroma-rich, basal/squamous, and neuroendocrine-like [23].

Luminal cancers arise in the superficial layer of the urothelium, and they show up-regulation of PARγ target genes with enrichment of FGFR3, ELF3, CDKN1A, and TSC1 mutations. However, basal tumors arise in deeper parts of the urothelium and show up-regulation of p63 target genes with enrichment of TP53 and RB1 mutations. It is also interesting that both types exhibit similarities to the basal and luminal subtypes identified in breast cancers. The recognition of those variants with an estimation of expression of only two markers, GATA3 (for luminal subtype) and KRT5/6 (for basal), is important because both types show different clinical outcomes and lead to different therapeutic approaches. Both types show different responses to frontline chemotherapy, with the basal subtype exhibiting more aggressive behavior with shorter survival than luminal cancers, but with much better response to platinum-based chemotherapy [24].

Proteins, which are synthesized from mutated genes and are overexpressed in luminal and basal subtypes of molecular subtypes of bladder cancer, may represent attractive therapeutic targets. They include E-Cadherin, HER2/3, Rab-25, and Src in luminal tumors and CD49, Cyclin B1, and FGFR in basal ones. Studies are ongoing on the use of specific agents personalized to the type of mutation present in the tumor cell. FGFRs are a family of receptor tyrosine kinases that are involved in tumor cell differentiation, proliferation, and angiogenesis. They can be upregulated in various tumor cell types, including bladder cancer. Erdafitinib, an oral pan-fibroblast growth factor receptor (FGFR) tyrosine kinase inhibitor, was approved for second-line treatment of patients who progressed following platinum-based chemotherapy. Its use is currently under investigation, and future implementation in bladder cancer treatment will be significant.

Another way to personalize treatment would be to incorporate modern immunotherapy into the clinical setting, possibly combining it with cisplatin chemotherapy. These include atezolizumab (blocking anti-PDL1 antibody) [25] and atezolizumab in combination with [26]. However, the most desirable would be a combination of pre-operative immunotherapy (possibly combined with chemotherapy) and its linking with surgery.

New discoveries in the field of molecular bladder cancer subtypes can improve treatment pathways of bladder cancer patients, not only in the advanced urothelial disease, but also in non-muscle invasive disease initially treated with transurethral resections and BCG installations. In the near future, various anti-tumor agents will be added to the treatment (currently, pembrolizumab studies are most promising) [27,28,29].

Clinical use of molecular subtyping is under investigation, so the decision about the type of treatment in patients with advanced urothelial carcinoma is currently based on clinical data only. For this purpose, it is necessary to divide patients into risk groups, using the most reliable and safe method possible. For clinical assessment, designated scales have been developed, which divide patients into those with moderate, high, and very high risk of post-operative complications.

Older adults generally have a worse prognosis and a higher postoperative complication rate. Therefore, radical treatment was proposed only to a small group of patients, and for the rest, only palliative treatment was used [4]. Currently, it is known that radical treatment in older patients can be safe when the type of treatment is personally adjusted to their state. With precise preoperative preparation, meticulous intraoperative hemostasis, a minimally invasive method of surgery (laparoscopic or robot-assisted), and the correct post-operative management, it is possible to significantly reduce surgery complication rates. In our previous study, older age alone did not increase the risk of major postoperative complications, prolonged hospital stays, or blood loss among patients who were subjected to radical cystectomy [30]. This concurs with other studies [31,32,33]. 

In the current study, we aimed to establish whether the personalized approach based on assessing risk with designed scales can be used, and whether the risk of complications can be determined before surgery. We also aimed to assess which of the four scales is best in terms of predicting major complications after radical cystectomy. It is interesting that the highest number of patients (58%) were screened as ECOG 0–1 and 35.2% as ASA 1–2. Self-reported results of the ECOG scale tended to present patients in better condition than the real situation, as seen by anesthesiologists (with use of the ASA scale). The G-8 is a short report scale, which should be used as a screening tool to identify patients in need of further assessment and appropriate geriatric assessment, with a threshold value of 14. Patients with a total score of 14 or lower are considered frail, and their radical treatment should be reconsidered. Indeed, in the group of patients with major complications, and in those who died, there were significantly more patients rated as frail [6]. On the other hand, the reduced complication rates in older patients in more recent studies might be explained by the greater availability of minimally invasive operations and better perioperative management.

In our study, the complication rate depended on the type of urinary derivation, which concurs with reports from other authors. In contrast, the presence of diabetes, renal insufficiency class III or more, and lower levels of albumins did not have a negative prognostic value in terms of the complication rate (in relation to urinary derivation). This proves the importance of good nutritional preparation before surgery as well as controlled diabetes. In our previous studies, we have shown that the type of approach had an impact on complication rates, with a shift towards more frequent, higher-grade complications in the open approach than the minimally invasive ones (laparoscopic and robot assisted) [34]. It was also demonstrated that the type of approach (ORC, LRC or RARC) had no impact on oncological safety in terms of positive surgical margins or number of lymph nodes removed. The significance of the PSM ratio was interestingly revealed in an article by Bongiolatti et al. [35].

Modern minimally invasive methods of approach (LRC or RARC) reduce the number of minor and hemorrhagic complications [36]. However, the type of approach must be adapted to the specific physiological limitations of the patient. Laparoscopic CO_2_ retention and hypoventilation with decreased venous return can result in metabolic acidosis. Additionally, the Trendelenburg position with peritoneal gas insufflations increases the intraabdominal pressure and can cause increased intracranial pressure during lower abdominal procedures. Therefore, laparoscopy in elderly adults was contraindicated because of pneumoperitoneum in cardiac and pulmonary physiology. With improved anesthetic and operating techniques, these contraindications are not valid in most cases. Moreover, several studies have confirmed the benefits of laparoscopy in elderly adults, including reduced hospital stays, lower rates of pulmonary complications, and fewer wound-healing problems compared to traditional operative approaches [37,38].

The use of LRC and RARC also decreases complication rates in terms of surgical site infections [39]. 

To conclude, evaluation of the patient’s frailty level and personalized geriatric assessment can lower the risk of complications.

## 5. Conclusions

It is possible to safely modify the treatment of patients with advanced neoplasmatic disease even when the 10-year survival rate is very low. In our study group, even in older and frail patients, the risk of fatal complications was low, probably due to the use of a minimally invasive surgical approach with short operation time and low blood loss. The patient’s general status should be assessed before the start of therapy because patients with a high risk of death or serious complications (evaluated with any rating scale) should be offered conservative treatment.

In a clinical setting, it seems that none of the scales can ascertain the risk of cystectomy because the percentage of patients with major complications among those who achieved worse score results on any scale is not different from the percentage of patients with major complications in the general group (Table 2). 

When more complex urinary derivation is applied, a significantly higher percentage of patients suffer higher grades of complications, so the patient’s general status should be assessed to personalize the type of urinary derivation. 

Precise pre-operative preparation of patients results in a significant reduction of complications; well-prepared patients in our study group did not show any differences in complication rates in relation to hypoalbuminemia, uncontrolled diabetes, or renal insufficiency.

## Figures and Tables

**Figure 1 jpm-12-00281-f001:**
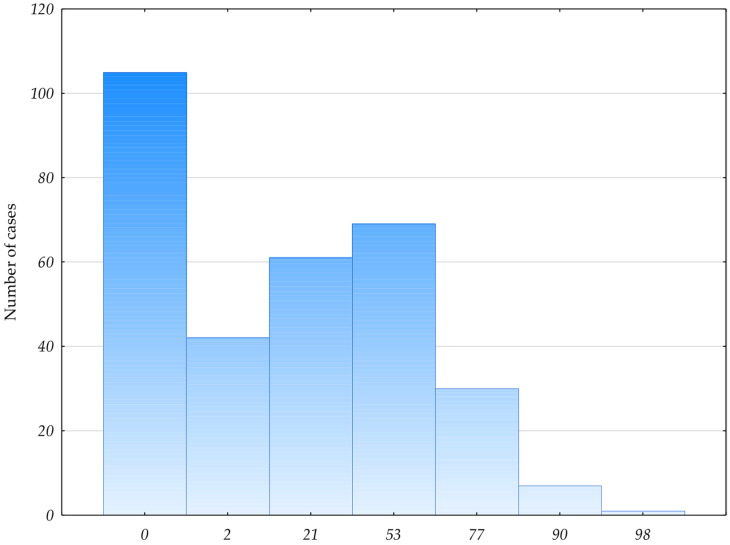
Chance of 10-year survival in the studied group according to the CCI scale. Number of cases: Number of patients with the indicated chance of 10-year survival. Chance expressed as a percentage.

**Figure 2 jpm-12-00281-f002:**
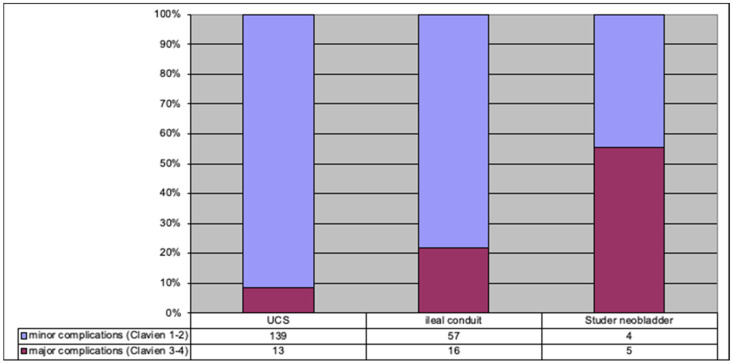
Complications in the study group according to Clavien–Dindo grades (1,2—minor complications, 3,4—major complications) depending on the urinary derivation.

**Figure 3 jpm-12-00281-f003:**
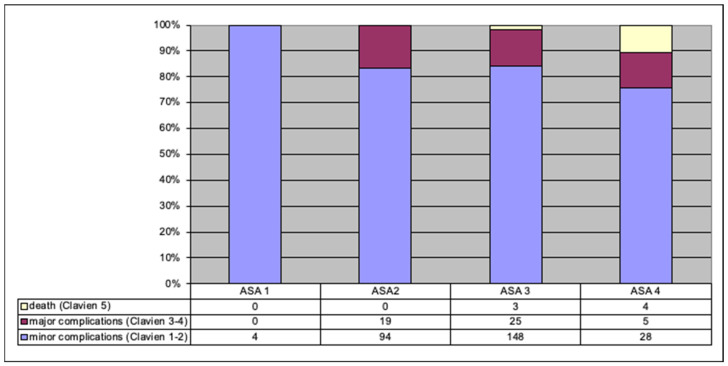
Relationship between ASA-assessed preoperative status and complications (number and percentage of patients).

**Figure 4 jpm-12-00281-f004:**
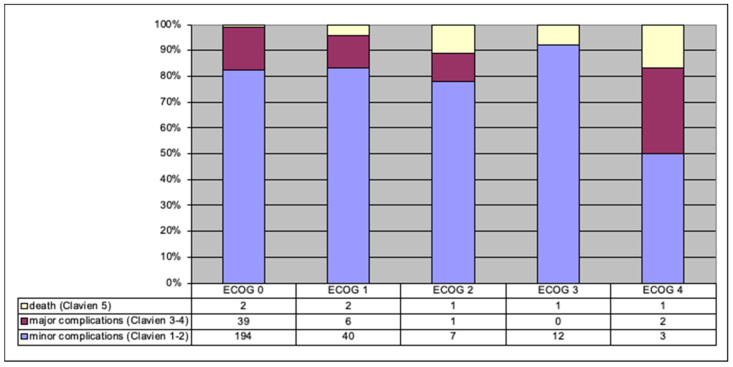
Relationship between ECOG-assessed preoperative status and complications (number and percentage of patients).

**Figure 5 jpm-12-00281-f005:**
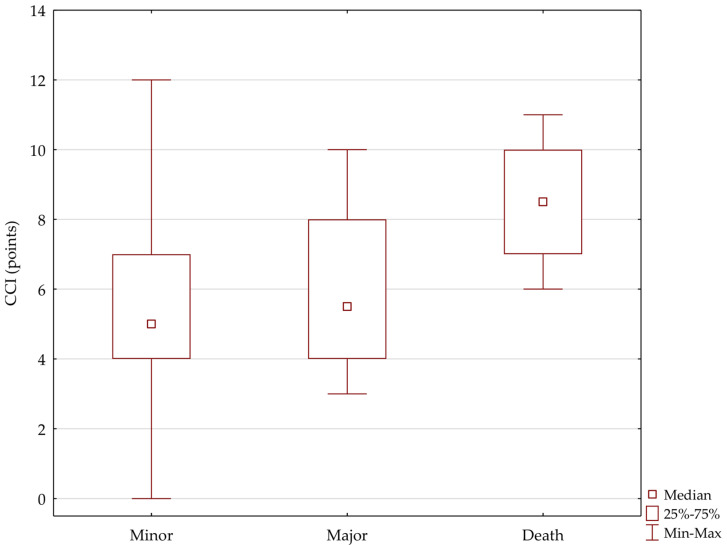
Relationship between CCI-assessed preoperative status and complications (number and percentage of patients).

**Figure 6 jpm-12-00281-f006:**
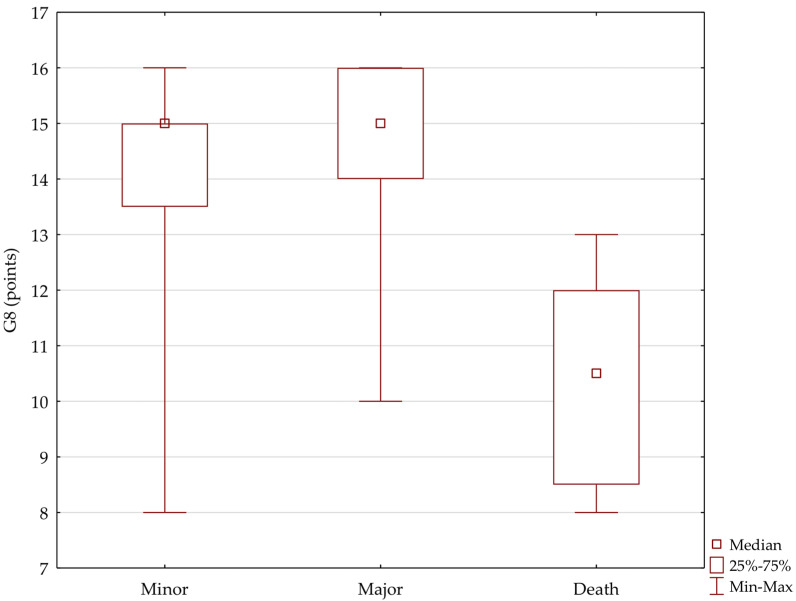
Relationship between G8-assessed preoperative status and complications (number and percentage of patients).

**Table 1 jpm-12-00281-t001:** Percentage of patients scored according to the appropriate scale.

ASA		ECOG		G8		CCI	
Score	% of Patients	Score	% of Patients	Score	% of Patients	Score	% of Patients
1	1	0	71	14–17	75	≥5	63
2	34	1	15	10–13	22	<5	32
3	54	2	3	0–9	3		
4	11	3	4				
		4	2				
		No Data	5			No Data	5

ASA-American Society of Anesthesiologists scale; CCI-Charlson Comorbidity Index; ECOG-Eastern Cooperative Oncology Group scale; G8-Geriatric-8 scale.

**Table 2 jpm-12-00281-t002:** Percentage of patients with a worse score according to the appropriate scale with major complications.

		ASA ≥ 3		ECOG ≥ 3		G8 ≤ 14		CCI ≥ 5		General
	N	%	N	%	N	%	N	%	N	%
Clavien 3–5	37/215	17	4/19	17	26/152	17	42/209	20	55/331	17

## Data Availability

The data presented in this study are available on request from the corresponding author.

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
