# Peer review of "A Personalized Approach to Radical Cystectomy Can Decrease Its Complication Rates"

_jpm, 2022, doi:10.3390/jpm12020281_

Round 1

Reviewer 1 Report

In this study, the authors analyzed the influence of a patient’s general status on perioperative morbidity and mortality after radical cystectomy. The authors studied which scale of patient’s general status evaluation (ASA, CCI, ECOG and G-8 scales) could be the best predictor of perioperative major complications.

In this series, the poor general condition of the patient is significantly correlated with a higher rate of post-operative complications. For this reason, the patient’s general status should be assessed to personalize the type of treatment and urinary derivation. However,

none of these scales has shown superiority.

In my opinion, this manuscript represents an original paper of up-to-date interest. The personalized approach to the patient is of great relevance and interest.

The manuscript is well structured and it could be suitable for publication after some

revisions in order to improve it:

- Please specify the average age of the patients evaluated

- It would be interesting to evaluate complication rates also based on the type of surgery performed (ORC, LCR and RARC) 

- Please state the importance of clear surgical margins whatever surgery performed (doi: 10.21037/jtd.2018.07.21)

- As partially stated in the text of the manuscript, research on genetics and miRNAs could also improve the therapeutic strategy and not only an early diagnosis and prognosis (doi:10.1016/j.urolonc.2017.08.004;   doi: 10.1016/j.urolonc.2021.11.001). Please, report the knowledge on the new future target therapy for bladder cancer (doi:10.1038/nrurol.2017.179)

- As mentioned in the text of the manuscript, the use of mini-invasive surgery can reduce

complication rates. Please improve knowledge about this topic (doi: 10.1515/med-2019-0081)

- English language should be revised by a native speaker

Author Response

Dear Reviewer.

Thank You very much for Your work and time You have spent with our article. With Your help and corrections I believe, it would be more interesting and descriptive to the reader.

Article reflects our own work, and difficulties we face every day when facing patient with advanced urothelial neoplasm. More we work with such patients, more questions are coming , and sometimes it is difficult to find answer. And this article is aiming to give us answers to the questions raised in everyday practice.

Below, please find answers, to Your questions and comments.

With best regards, on behalf of authors

In this study, the authors analyzed the influence of a patient’s general status on perioperative morbidity and mortality after radical cystectomy. The authors studied which scale of patient’s general status evaluation (ASA, CCI, ECOG and G-8 scales) could be the best predictor of perioperative major complications.

In this series, the poor general condition of the patient is significantly correlated with a higher rate of post-operative complications. For this reason, the patient’s general status should be assessed to personalize the type of treatment and urinary derivation. However,

none of these scales has shown superiority.

In my opinion, this manuscript represents an original paper of up-to-date interest. The personalized approach to the patient is of great relevance and interest.

The manuscript is well structured and it could be suitable for publication after some

revisions in order to improve it:

  1. Please specify the average age of the patients evaluated

Thank You for question- the average age of patient is 68,5 years.

W have added this information to the manuscript.

  1. It would be interesting to evaluate complication rates also based on the type of surgery performed (ORC, LCR and RARC) 

Thank You for Your comment. Such evaluation was performed before in our previous paper (please find it here: Urol Int 2022;106:163–170 (DOI:10.1159/000517787), where we state that: "The type of surgical approach was significantly related to the number of perioperative complications (p < 0.00005). The majority of LRC and RARC operations resulted in Grade 1 complications  (52.5% and 51.1%, respectively), while Grade 2 complications were observed more frequently with ORC (78.6% of cases). Furthermore, less severe complications (i.e., Grade 1–2 vs. Grade 2–3) were observed in patients who underwent LRC and RARC with UCS or ileal conduit than those in the ORC group (p = 0.0012 for UCS, p < 0.00005 for ileal conduit, and p < 0.00005 for orthotropic neobladder)."

This was study performed in another group of patients, much bigger than current (533 patients) operated with three mentioned surgical approaches. Type of approach had impact on complication rate, and lower grade of them were more frequent in LRC and RARC, than ORC. The same applied to the group of patient with ileal conduit and simple ureterocutaneostomy in relation to the type of surgical approach. Since this was not aim of the study to compare three methods, and it was previously done in bigger group, we decided not to include this in our analysis.

Sentence was added to discussion part of article:

“In our previous studies we have proved, that type of the approach had impact on complication rates, with shift towards more frequent higher grade complications in open approach, than minimally invasive (laparoscopic and robot assisted).

  1. Please state the importance of clear surgical margins whatever surgery performed (doi: 10.21037/jtd.2018.07.21)

Positive surgical margins rate are crucial in terms of oncologic safety. In our previous studies we have proved, that PSM ratio was not affected by the type of approach.  Therefore it can be assumed, that robot assisted and laparoscopic approach was as same safe in term of oncologic results as classical open approach. On the other hand, I also believe, that it is true not in every urological clinic, especially this, which is starting minimally invasive approach to the cystectomy. Radical cystectomy is difficult and demanding procedure. It requires high laparoscopic skills, and learning curve (especially in LRC) is long. In our department laparoscopic procedures are performed for long time, and radical prostatectomy was introduced more than 15 years ago, therefore shift from open to laparoscopic cystectomy was not that difficult. When robot assisted approach was introduced in our department, we were already skilled in laparoscopic cystectomy, therefore in was not as difficult as it could be. It was interesting for us to compare results of our surgery, therefore we have compared results of open and laparoscopic and robot assisted approach it terms of complication rate and oncologic ssaftety (PSM ratio was one of them). And we have seen not significant differences between groups, so in our ase it can be assumed that type of approach does not matter, but other factors may play its role, like age, time from last TURBT to the cystectomy or of course not surprisingly, pathological stage of disease.

So sentence was added to discussion part of article, which was following sentence above (in answer number 3), and reference:

It was also proved, that type of the approach (ORC, LRC or RARC) had no impact on oncological safety in terms of positive surgical margins or number of lyphnodes removed. Significance of PSM ratio was interestingly revealed in article by Bongiolatti et al. [Bongiolatti S, Corzani R, Borgianni S, Meniconi F, Cipollini F, Gonfiotti A, Viggiano D, Paladini P, Voltolini L. Long-term results after surgical treatment of the dominant lung adenocarcinoma associated with ground-glass opacities. J Thorac Dis. 2018 Aug;10(8):4838-4848. doi: 10.21037/jtd.2018.07.21. PMID: 30233857; PMCID: PMC6129865.]

  1. As partially stated in the text of the manuscript, research on genetics and miRNAs could also improve the therapeutic strategy and not only an early diagnosis and prognosis (doi:10.1016/j.urolonc.2017.08.004;   doi: 10.1016/j.urolonc.2021.11.001). Please, report the knowledge on the new future target therapy for bladder cancer (doi:10.1038/nrurol.2017.179)

Thank You for Your comment. In discussion part of the article it was briefly mentioned future possible pathways to improve oncological results of treatment of patients with advanced urothelial cancer. In fact, miRNAs and other genetic based assays could be helpful in distinguishing right patient and to provide best treatment. In current situation we put all patient into the same “basket”, and probably treat different types of the disease with the same approach. The same was applied to breast cancer, where treatment was totally rearranged according to the  recent genetics and molecular discoveries. The same will surely will be applied to the bladder cancer. One possible way of course will be use of genomics proteomics and treatment based on miRNAs.

Therefore sentence was added to the discussion part of the article, and references:

“New discoveries in field of molecular bladder cancer subtypes can improve treatment pathways of bladder cancer patients, not only in the advanced urothelial disease, but also in non muscle invasive disease initially treated with transurethral resections and BCG installations. In close future various anti-tumor agents will be added to the treatment, (currently pembrolizumab studies are most promising). [Giulia Poli, Giovanni Cochetti, Andrea Boni, Maria Giulia Egidi, Stefano Brancorsini, Ettore Mearini, Characterization of inflammasome-related genes in urine sediments of patients receiving intravesical BCG therapy, Urologic Oncology: Seminars and Original Investigations, Volume 35, Issue 12, 2017, Pages 674.e19-674.e24, DOI: 10.1016/j.urolonc.2017.08.004.]. [Cochetti G, Rossi de Vermandois JA, Maulà V, Cari L, Cagnani R, Suvieri C, Balducci PM, Paladini A, Del Zingaro M, Nocentini G, Mearini E. Diagnostic performance of the Bladder EpiCheck methylation test and photodynamic diagnosis-guided cystoscopy in the surveillance of high-risk non-muscle invasive bladder cancer: A single centre, prospective, blinded clinical trial. Urol Oncol. 2021 Dec 12:S1078-1439(21)00483-X. doi: 10.1016/j.urolonc.2021.11.001. Epub ahead of print. PMID: 34911649.] Felsenstein, K., Theodorescu, D. Precision medicine for urothelial bladder cancer: update on tumour genomics and immunotherapy. Nat Rev Urol 15, 92–111 (2018). https://doi.org/10.1038/nrurol.2017.179

  1. As mentioned in the text of the manuscript, the use of mini-invasive surgery can reduce

complication rates. Please improve knowledge about this topic (doi: 10.1515/med-2019-0081)

Thank You for the comment. It is true, we see, that use of minimally invasive surgery reduces complication rate, also surgical site infection.

Therefore sentence was added to the discussion part, and reference: :

"Use of LRC and RARC decreases complication rate also in terms of surgical site infection. [de Vermandois JAR, Cochetti G, Zingaro MD, Santoro A, Panciarola M, Boni A, Marsico M, Gaudio G, Paladini A, Guiggi P, Cirocchi R, Mearini E. Evaluation of Surgical Site Infection in Mini-invasive Urological Surgery. Open Med (Wars). 2019 Sep 15;14:711-718. doi: 10.1515/med-2019-0081]."

  1. English language should be revised by a native speaker

Thank You for comment. English language was corrected by native speaker.

Reviewer 2 Report

General comment

In this paper, the authors assessed the influence of patients' general status on perioperative morbidity and mortality after radical cystectomy and the use of scales in terms of prediction of major complications. Data of 331 patients with muscle-invasive bladder cancer who underwent radical cystectomy was analyzed.

Overall, the topic is not original or timely. However, the paper is well-written and structured.

Some points are argued below.

Major corrections

-Introduction. Please the authors reduce the introduction. It seems to belong.

-Introduction. Please the authors better address the sentence “Radical treatment of urothelial cancer patients consists of neoadjuvant or adjuvant chemotherapy, followed by radical surgery” (lines 53-54, page 5) as the sentence appears to not indicate surgical stage. Did the authors reference specific urothelial cancer or specific stage? Please add the reference.

-Introduction. Please the authors review the percentages of As much as 75% of these cancers are classified as pure urothelial carcinoma & histological variants. It seems to not reflect the current literature.

-Material and methods. Very well done. Please specify that complications were stratify according to urinary deviation.

-Material and methods. Please the authors better address if they used Statistical significance as p ≤ 0.05 or p<0.05.

-Resuts. Table with general patient information is required. Granular patients information lack. They can enrich the results and discussion.

-Discussion. Like the introduction, it seems to be very long. Please consider reducing it.

Minor corrections

Please better address Figure 1. It seems to be unclear.

Please add a reference at line 65 page 5.

Please add a reference at line 229 page 5.

Grammar & Language double-check is mandatory

Author Response

Dear Reviewer.

Thank You very much for Your work and time You have spent with our article. With Your help and corrections I believe, it would be more interesting and descriptive to the reader.

Article reflects our own work, and difficulties we face every day when facing patient with advanced urothelial neoplasm. More we work with such patients, more questions are coming , and sometimes it is difficult to find answer. And this article is aiming to give us answers to the questions raised in everyday practice.

Below, please find answers, to Your questions and comments.

With best regards, on behalf of authors

General comment

In this paper, the authors assessed the influence of patients' general status on perioperative morbidity and mortality after radical cystectomy and the use of scales in terms of prediction of major complications. Data of 331 patients with muscle-invasive bladder cancer who underwent radical cystectomy was analyzed.

Overall, the topic is not original or timely. However, the paper is well-written and structured.

Some points are argued below.

Major corrections

  1. Introduction. Please the authors reduce the introduction. It seems to belong.

Thank You for that comment. Introduction was reduced.

  1. Introduction. Please the authors better address the sentence “Radical treatment of urothelial cancer patients consists of neoadjuvant or adjuvant chemotherapy, followed by radical surgery” (lines 53-54, page 5) as the sentence appears to not indicate surgical stage. Did the authors reference specific urothelial cancer or specific stage? Please add the reference.

Thank You for that comment. In fact, sentence was not clear. It was changed to:

"Radical treatment of locally advanced urothelial cancer patients consists of neoadjuvant chemotherapy, followed by radical surgery"

  1. Introduction. Please the authors review the percentages of As much as 75% of these cancers are classified as pure urothelial carcinoma & histological variants. It seems to not reflect the current literature.

Thank You for that remark. It is true, it should not be written in such way. Therefore sentence was changed to:

"Majority of these cancers are classified as pure urothelial carcinoma, and rest are assessed as other histological variants."

  1. Material and methods. Very well done. Please specify that complications were stratify according to urinary deviation.

A sentence which specified that complications were stratify according to urinary deviation was added.

  1. Material and methods. Please the authors better address if they used Statistical significance as p ≤ 0.05 or p<0.05.

Thank You for this comment. It was changed to p<0.05.

  1. Results. Table with general patient information is required. Granular patients information lack. They can enrich the results and discussion.

Thank You for Your comment. Such information was added to the Material part of manuscript.

  1. Discussion. Like the introduction, it seems to be very long. Please consider reducing it.

Discussion section was also reduced.

Minor corrections

  1. Please better address Figure 1. It seems to be unclear.

It was renamed, according to Your suggestion

  1. Please add a reference at line 65 page 5.

Thank You for comment. Reference was added

  1. Please add a reference at line 229 page 5.

Thank You for the comment. Reference was added

  1. Grammar & Language double-check is mandatory

Thank You for the comment. English language was corrected by native speaker.